# Maximum Likelihood Learning With Arbitrary Treewidth via Fast-Mixing Parameter Sets

**Justin Domke**
NICTA, Australian National University
`justin.domke@nicta.com.au`

## Abstract

Inference is typically intractable in high-treewidth undirected graphical models, making maximum likelihood learning a challenge. One way to overcome this is to restrict parameters to a tractable set, most typically the set of tree-structured parameters. This paper explores an alternative notion of a tractable set, namely a set of "fast-mixing parameters" where Markov chain Monte Carlo (MCMC) inference can be guaranteed to quickly converge to the stationary distribution. While it is common in practice to approximate the likelihood gradient using samples obtained from MCMC, such procedures lack theoretical guarantees. This paper proves that for any exponential family with bounded sufficient statistics, (not just graphical models) when parameters are constrained to a fast-mixing set, gradient descent with gradients approximated by sampling will approximate the maximum likelihood solution inside the set with high-probability. When unregularized, to find a solution $\epsilon$-accurate in log-likelihood requires a total amount of effort cubic in $1/\epsilon$, disregarding logarithmic factors. When ridge-regularized, strong convexity allows a solution $\epsilon$-accurate in parameter distance with effort quadratic in $1/\epsilon$. Both of these provide of a fully-polynomial time randomized approximation scheme.

## 1 Introduction

In undirected graphical models, maximum likelihood learning is intractable in general. For example, Jerrum and Sinclair [1993] show that evaluation of the partition function (which can easily be computed from the likelihood) for an Ising model is #P-complete, and that even the existence of a fully-polynomial time randomized approximation scheme (FPRAS) for the partition function would imply that RP = NP.

If the model is well-specified (meaning that the target distribution falls in the assumed family) then there exist several methods that can efficiently recover correct parameters, among them the pseudolikelihood [3], score matching [16, 22], composite likelihoods [20, 30], Mizrahi et al.'s [2014] method based on parallel learning in local clusters of nodes and Abbeel et al.'s [2006] method based on matching local probabilities. While often useful, these methods have some drawbacks. First, these methods typically have inferior sample complexity to the likelihood. Second, these all assume a well-specified model. If the target distribution is not in the assumed class, the maximum-likelihood solution will converge to the M-projection (minimum of the KL-divergence), but these estimators do not have similar guarantees. Third, even when these methods succeed, they typically yield a distribution in which inference is still intractable, and so it may be infeasible to actually make use of the learned distribution.

Given these issues, a natural approach is to restrict the graphical model parameters to a tractable set $\Theta$, in which learning and inference can be performed efficiently. The gradient of the likelihood is determined by the marginal distributions, whose difficulty is typically determined by the treewidth of the graph. Thus, probably the most natural tractable family is the set of tree-structured distributions,

where $\Theta = \{\theta : \exists \text{tree } T, \forall (i,j) \notin T, \theta_{ij} = 0\}$. The Chow-Liu algorithm [1968] provides an efficient method for finding the maximum likelihood parameter vector $\theta$ in this set, by computing the mutual information of all empirical pairwise marginals, and finding the maximum spanning tree. Similarly, Heinemann and Globerson [2014] give a method to efficiently learn high-girth models where correlation decay limits the error of approximate inference, though this will not converge to the M-projection when the model is mis-specified.

This paper considers a fundamentally different notion of tractability, namely a guarantee that Markov chain Monte Carlo (MCMC) sampling will quickly converge to the stationary distribution. Our fundamental result is that if $\Theta$ is such a set, and one can project onto $\Theta$, then there exists a FPRAS for the maximum likelihood solution inside $\Theta$. While inspired by graphical models, this result works entirely in the exponential family framework, and applies generally to any exponential family with bounded sufficient statistics.

The existence of a FPRAS is established by analyzing a common existing strategy for maximum likelihood learning of exponential families, namely gradient descent where MCMC is used to generate samples and approximate the gradient. It is natural to conjecture that, if the Markov chain is fast mixing, is run long enough, and enough gradient descent iterations are used, this will converge to nearly the optimum of the likelihood inside $\Theta$, with high probability. This paper shows that this is indeed the case. A separate analysis is used for the ridge-regularized case (using strong convexity) and the unregularized case (which is merely convex).

## 2 Setup

Though notation is introduced when first used, the most important symbols are given here for more reference.

- $\theta$ - parameter vector to be learned
- $\mathbb{M}_\theta$ - Markov chain operator corresponding to $\theta$
- $\theta_k$ - estimated parameter vector at $k$-th gradient descent iteration
- $q_k = \mathbb{M}_{\theta_{k-1}}^v r$ - approximate distribution sampled from at iteration $k$. ($v$ iterations of the Markov chain corresponding to $\theta_{k-1}$ from arbitrary starting distribution $r$.)
- $\Theta$ - constraint set for $\theta$
- $f$ - negative log-likelihood on training data
- $L$ - Lipschitz constant for the gradient of $f$.
- $\theta^* = \arg\min_{\theta \in \Theta} f(\theta)$ - minimizer of likelihood inside of $\Theta$
- $K$ - total number of gradient descent steps
- $M$ - total number of samples drawn via MCMC
- $N$ - length of vector $x$.
- $v$ - number of Markov chain transitions applied for each sample
- $C, \alpha$ - parameters determining the mixing rate of the Markov chain. (Equation 3)
- $R_a$ - sufficient statistics norm bound.
- $\epsilon_f$ - desired optimization accuracy for $f$
- $\epsilon_\theta$ - desired optimization accuracy for $\theta$
- $\delta$ - permitted probability of failure to achieve a given approximation accuracy

This paper is concerned with an exponential family of the form

$$p_\theta(x) = \exp(\theta \cdot t(x) - A(\theta)),$$

where $t(x)$ is a vector of sufficient statistics, and the log-partition function $A(\theta)$ ensures normalization. An undirected model can be seen as an exponential family where $t$ consists of indicator functions for each possible configuration of each clique [32]. While such graphical models motivate this work, the results are most naturally stated in terms of an exponential family and apply more generally.

- Initialize $\theta_0 = 0$.
- For $k = 1, 2, ..., K$
  - Draw samples. For $i = 1, ..., M$, sample
    $$x_i^{k-1} \sim q_{k-1} := \mathbb{M}_{\theta_{k-1}}^v r.$$
  - Estimate the gradient as
    $$f'(\theta_{k-1}) + e_k \leftarrow \frac{1}{M} \sum_{i=1}^{M} t(x_i^{k-1}) - \bar{t} + \lambda\theta.$$
  - Update the parameter vector as
    $$\theta_k \leftarrow \Pi_\Theta \left[ \theta_{k-1} - \frac{1}{L}\left(f'(\theta_{k-1}) + e_k)\right) \right].$$
- Output $\theta_K$ or $\frac{1}{K}\sum_{k=1}^{K} \theta_k$.

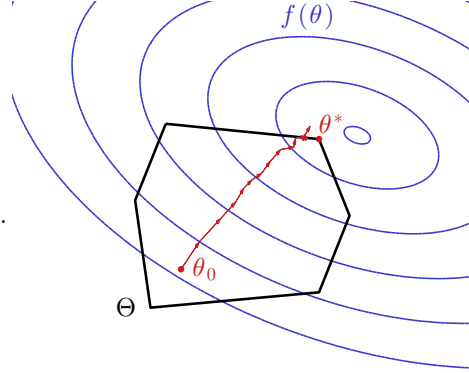

Figure 1: Left: **Algorithm 1**, approximate gradient descent with gradients approximated via MCMC, analyzed in this paper. Right: A cartoon of the desired performance, stochastically finding a solution near $\theta^*$, the minimum of the regularized negative log-likelihood $f(\theta)$ in the set $\Theta$.

We are interested in performing maximum-likelihood learning, i.e. minimizing, for a dataset $z_1, ..., z_D$,

$$f(\theta) = -\frac{1}{D}\sum_{i=1}^{D} \log p_\theta(z_i) + \frac{\lambda}{2}\|\theta\|_2^2 = A(\theta) - \theta \cdot \bar{t} + \frac{\lambda}{2}\|\theta\|_2^2, \tag{1}$$

where we define $\bar{t} = \frac{1}{D}\sum_{i=1}^{D} t(z_i)$. It is easy to see that the gradient of $f$ takes the form

$$f'(\theta) = \mathbb{E}_{p_\theta}[t(X)] - \bar{t} + \lambda\theta.$$

If one would like to optimize $f$ using a gradient-based method, computing the expectation of $t(X)$ with respect to $p_\theta$ can present a computational challenge. With discrete graphical models, the expected value of $t$ is determined by the marginal distributions of each factor in the graph. Typically, the computational difficulty of computing these marginal distributions is determined by the treewidth of the graph– if the graph is a tree, (or close to a tree) the marginals can be computed by the junction-tree algorithm [18]. One option, with high treewidth, is to approximate the marginals with a variational method. This can be seen as exactly optimizing a "surrogate likelihood" approximation of Eq. 1 [31].

Another common approach is to use Markov chain Monte Carlo (MCMC) to compute a sample $\{x_i\}_{i=1}^{M}$ from a distribution close to $p_\theta$, and then approximate $\mathbb{E}_{p_\theta}[t(X)]$ by $(1/M)\sum_{i=1}^{M} t(x_i)$. This strategy is widely used, varying in the model type, the sampling algorithm, how samples are initialized, the details of optimization, and so on [10, 25, 27, 24, 7, 33, 11, 2, 29, 5]. Recently, Steinhardt and Liang [28] proposed learning in terms of the stationary distribution obtained from a chain with a nonzero restart probability, which is fast-mixing by design.

While popular, such strategies generally lack theoretical guarantees. If one were able to exactly sample from $p_\theta$, this could be understood simply as stochastic gradient descent. But, with MCMC, one can only sample from a distribution approximating $p_\theta$, meaning the gradient estimate is not only noisy, but also biased. In general, one can ask how should the step size, number of iterations, number of samples, and number of Markov chain transitions be set to achieve a convergence level.

The gradient descent strategy analyzed in this paper, in which one updates a parameter vector $\theta_k$ using approximate gradients is outlined and shown as a cartoon in Figure 1. Here, and in the rest of the paper, we use $p_k$ as a shorthand for $p_{\theta_k}$, and we let $e_k$ denote the difference between the estimated gradient and the true gradient $f'(\theta_{k-1})$. The projection operator is defined by $\Pi_\Theta[\phi] = \arg\min_{\theta \in \Theta}\|\theta - \phi\|_2$.

We assume that the parameter set $\theta$ is constrained to a set $\Theta$ such that MCMC is guaranteed to mix at a certain rate (Section 3.1). With convexity, this assumption can bound the mean and variance

of the errors at each iteration, leading to a bound on the sum of errors. With strong convexity, the error of the gradient at each iteration is bounded with high probability. Then, using results due to [26] for projected gradient descent with errors in the gradient, we show a schedule the number of iterations $K$, the number of samples $M$, and the number of Markov transitions $v$ such that with high probability,

$$f\left(\frac{1}{K}\sum_{k=1}^{K}\theta_k\right) - f\left(\theta^*\right) \leq \epsilon_f \text{ or } \|\theta_K - \theta^*\|_2 \leq \epsilon_\theta,$$

for the convex or strongly convex cases, respectively, where $\theta^* \in \arg\min_{\theta\in\Theta} f(\theta)$. The total number of Markov transitions applied through the entire algorithm, $KMv$ grows as $(1/\epsilon_f)^3 \log(1/\epsilon_f)$ for the convex case, $(1/\epsilon_\theta^2)\log(1/\epsilon_\theta^2)$ for the strongly convex case, and polynomially in all other parameters of the problem.

## 3 Background

### 3.1 Mixing times and Fast-Mixing Parameter Sets

This Section discusses some background on mixing times for MCMC. Typically, mixing times are defined in terms of the **total-variation distance** $\|p-q\|_{TV} = \max_A |p(A) - q(A)|$, where the maximum ranges over the sample space. For discrete distributions, this can be shown to be equivalent to $\|p-q\|_{TV} = \frac{1}{2}\sum_x |p(x) - q(x)|$.

We assume that a sampling algorithm is known, a single iteration of which can be thought of an operator $\mathbb{M}_\theta$ that transforms some starting distribution into another. The stationary distribution is $p_\theta$, i.e. $\lim_{v\to\infty} \mathbb{M}_\theta^v q = p_\theta$ for all $q$. Informally, a Markov chain will be fast mixing if the total variation distance between the starting distribution and the stationary distribution decays rapidly in the length of the chain. This paper assumes that a convex set $\Theta$ and constants $C$ and $\alpha$ are known such that for all $\theta \in \Theta$ and all distributions $q$,

$$\|\mathbb{M}_\theta^v q - p_\theta\|_{TV} \leq C\alpha^v. \tag{2}$$

This means that the distance between an arbitrary starting distribution $q$ and the stationary distribution $p_\theta$ decays geometrically in terms of the number of Markov iterations $v$. This assumption is justified by the Convergence Theorem [19, Theorem 4.9], which states that if $\mathbb{M}$ is irreducible and aperiodic with stationary distribution $p$, then there exists constants $\alpha \in (0,1)$ and $C > 0$ such that

$$d(v) := \sup_q \|\mathbb{M}^v q - p\|_{TV} \leq C\alpha^v. \tag{3}$$

Many results on mixing times in the literature, however, are stated in a less direct form. Given a constant $\epsilon$, the **mixing time** is defined by $\tau(\epsilon) = \min\{v : d(v) \leq \epsilon\}$. It often happens that bounds on mixing times are stated as something like $\tau(\epsilon) \leq \lceil a + b\ln\frac{1}{\epsilon}\rceil$ for some constants $a$ and $b$. It follows from this that $\|\mathbb{M}^v q - p\|_{TV} \leq C\alpha^v$ with $C = \exp(a/b)$ and $\alpha = \exp(-1/b)$.

A simple example of a fast-mixing exponential family is the Ising model, defined for $x \in \{-1, +1\}^N$ as

$$p(x|\theta) = \exp\left(\sum_{(i,j)\in\text{Pairs}} \theta_{ij} x_i x_j + \sum_i \theta_i x_i - A(\theta)\right).$$

A simple result for this model is that, if the maximum degree of any node is $\Delta$ and $|\theta_{ij}| \leq \beta$ for all $(i,j)$, then for univariate Gibbs sampling with random updates, $\tau(\epsilon) \leq \lceil\frac{N\log(N/\epsilon)}{1-\Delta\tanh(\beta)}\rceil$ [19]. The algorithm discussed in this paper needs the ability to project some parameter vector $\phi$ onto $\Theta$ to find $\arg\min_{\theta\in\Theta}\|\theta-\phi\|_2$. Projecting a set of arbitrary parameters onto this set of fast-mixing parameters is trivial– simply set $\theta_{ij} = \beta$ for $\theta_{ij} > \beta$ and $\theta_{ij} \leftarrow -\beta$ for $\theta_{ij} < -\beta$.

For more dense graphs, it is known [12, 9] that, for a matrix norm $\|\cdot\|$ that is the spectral norm $\|\cdot\|_2$, or induced 1 or infinity norms,

$$\tau(\epsilon) \leq \left\lceil\frac{N\log(N/\epsilon)}{1-\|R(\theta)\|}\right\rceil \tag{4}$$

where $R_{ij}(\theta) = |\theta_{ij}|$. Domke and Liu [2013] show how to perform this projection for the Ising model when $\|\cdot\|$ is the spectral norm $\|\cdot\|_2$ with a convex optimization utilizing the singular value decomposition in each iteration.

Loosely speaking, the above result shows that univariate Gibbs sampling on the Ising model is fast-mixing, as long as the interaction strengths are not too strong. Conversely, Jerrum and Sinclair [1993] exhibited an alternative Markov chain for the Ising model that is rapidly mixing for *arbitrary* interaction strengths, provided the model is ferromagnetic, i.e. that all interaction strengths are positive with $\theta_{ij} \geq 0$ and that the field is unidirectional. This Markov chain is based on sampling in different "subgraphs world" state-space. Nevertheless, it can be used to estimate derivatives of the Ising model log-partition function with respect to parameters, which allows estimation of the gradient of the log-likelihood. Huber [2012] provided a simulation reduction to obtain an Ising model sample from a subgraphs world sample.

More generally, Liu and Domke [2014] consider a pairwise Markov random field, defined as

$$p(x|\theta) = \exp\left(\sum_{i,j} \theta_{ij}(x_i, x_j) + \sum_i \theta_i(x_i) - A(\theta)\right),$$

and show that, if one defines $R_{ij}(\theta) = \max_{a,b,c} \frac{1}{2}|\theta_{ij}(a,b) - \theta_{ij}(a,c)|$, then again Equation 4 holds. An algorithm for projecting onto the set $\Theta = \{\theta : \|R(\theta)\| \leq c\}$ exists.

There are many other mixing-time bounds for different algorithms, and different types of models [19]. The most common algorithms are univariate Gibbs sampling (often called Glauber dynamics in the mixing time literature) and Swendsen-Wang sampling. The Ising model and Potts models are the most common distributions studied, either with a grid or fully-connected graph structure. Often, the motivation for studying these systems is to understand physical systems, or to mathematically characterize phase-transitions in mixing time that occur as interactions strengths vary. As such, many existing bounds assume uniform interaction strengths. For all these reasons, these bounds typically require some adaptation for a learning setting.

## 4 Main Results

### 4.1 Lipschitz Gradient

For lack of space, detailed proofs are postponed to the appendix. However, informal proof sketches are provided to give some intuition for results that have longer proofs. Our first main result is that the regularized log-likelihood has a Lipschitz gradient.

**Theorem 1.** *The regularized log-likelihood gradient is L-Lipschitz with $L = 4R_2^2 + \lambda$, i.e.*

$$\|f'(\theta) - f'(\phi)\|_2 \leq (4R_2^2 + \lambda)\|\theta - \phi\|_2.$$

*Proof sketch.* It is easy, by the triangle inequality, that $\|f'(\theta) - f'(\phi)\|_2 \leq \|\frac{dA}{d\theta} - \frac{dA}{d\phi}\|_2 + \lambda\|\theta - \phi\|_2$. Next, using the assumption that $\|t(x)\|_2 \leq R_2$, one can bound that $\|\frac{dA}{d\theta} - \frac{dA}{d\phi}\|_2 \leq 2R_2\|p_\theta - p_\phi\|_{TV}$. Finally, some effort can bound that $\|p_\theta - p_\phi\|_{TV} \leq 2R_2\|\theta - \phi\|_2$. $\qquad\square$

### 4.2 Convex convergence

Now, our first major result is a guarantee on the convergence that is true both in the regularized case where $\lambda > 0$ and the unregularized case where $\lambda = 0$.

**Theorem 2.** *With probability at least $1 - \delta$, at long as $M \geq 3K/\log(\frac{1}{\delta})$, Algorithm 1 will satisfy*

$$f\left(\frac{1}{K}\sum_{k=1}^{K}\theta_k\right) - f(\theta^*) \leq \frac{8R_2^2}{KL}\left(\frac{L\|\theta_0 - \theta^*\|_2}{4R_2} + \log\frac{1}{\delta} + \frac{K}{\sqrt{M}} + KC\alpha^v\right)^2.$$

*Proof sketch.* First, note that $f$ is convex, since the Hessian of $f$ is the covariance of $t(X)$ when $\lambda = 0$ and $\lambda > 0$ only adds a quadratic. Now, define the quantity $d_k = \frac{1}{M}\sum_{m=1}^{M} t(X_m^k) - $

$\mathbb{E}_{q_k}[t(X)]$ to be the difference between the estimated expected value of $t(X)$ under $q_k$ and the true value. An elementary argument can bound the expected value of $\|d_k\|$, while the Efron-Stein inequality can bounds its variance. Using both of these bounds in Bernstein's inequality can then show that, with probability $1 - \delta$, $\sum_{k=1}^{K} \|d_k\| \leq 2R_2(K/\sqrt{M} + \log \frac{1}{\delta})$. Finally, we can observe that $\sum_{k=1}^{K} \|e_k\| \leq \sum_{k=1}^{K} \|d_k\| + \sum_{k=1}^{K} \|\mathbb{E}_{q_k}[t(X)] - \mathbb{E}_{p_{\theta_k}}[t(X)]\|_2$. By the assumption on mixing speed, the last term is bounded by $2KR_2C\alpha^v$. And so, with probability $1 - \delta$, $\sum_{k=1}^{K} \|e_k\| \leq 2R_2(K/\sqrt{M} + \log \frac{1}{\delta}) + 2KR_2C\alpha^v$. Finally, a result due to Schmidt et al. [26] on the convergence of gradient descent with errors in estimated gradients gives the result. $\quad\square$

Intuitively, this result has the right character. If $M$ grows on the order of $K^2$ and $v$ grows on the order of $\log K/(-\log \alpha)$, then all terms inside the quadratic will be held constant, and so if we set $K$ of the order $1/\epsilon$, the sub-optimality will on the order of $\epsilon$ with a total computational effort roughly on the order of $(1/\epsilon)^3 \log(1/\epsilon)$. The following results pursue this more carefully. Firstly, one can observe that a minimum amount of work must be performed.

**Theorem 3.** *For $a, b, c, \alpha > 0$, if $K, M, v > 0$ are set so that $\frac{1}{K}(a + b\frac{K}{\sqrt{M}} + Kc\alpha^v)^2 \leq \epsilon$, then*

$$KMv \geq \frac{a^4b^2}{\epsilon^3} \frac{\log \frac{ac}{\epsilon}}{(-\log \alpha)}.$$

*Since it must be true that $a/\sqrt{K} + b\sqrt{K/M} + \sqrt{K}c\alpha^v \leq \sqrt{\epsilon}$, each of these three terms must also be at most $\sqrt{\epsilon}$, giving lower-bounds on $K$, $M$, and $v$. Multiplying these gives the result.*

Next, an explicit schedule for $K$, $M$, and $v$ is possible, in terms of a convex set of parameters $\beta_1, \beta_2, \beta_3$. Comparing this to the lower-bound above shows that this is not too far from optimal.

**Theorem 4.** *Suppose that $a, b, c, \alpha > 0$. If $\beta_1 + \beta_2 + \beta_3 = 1$, $\beta_1, \beta_2, \beta_3 > 0$, then setting $K = \frac{a^2}{\beta_1^2 \epsilon}$, $M = (\frac{ab}{\beta_1\beta_2\epsilon})^2$, $v = \log \frac{ac}{\beta_1\beta_3\epsilon}/(-\log \alpha)$ is sufficient to guarantee that $\frac{1}{K}(a + b\frac{K}{\sqrt{M}} + Kc\alpha^v)^2 \leq \epsilon$ with a total work of*

$$KMv = \frac{1}{\beta_1^4\beta_2^2} \frac{a^4b^2}{\epsilon^3} \frac{\log \frac{ac}{\beta_1\beta_3\epsilon}}{(-\log \alpha)}.$$

*Simply verify that the $\epsilon$ bound holds, and multiply the terms together.*

For example, setting $\beta_1 = 0.66$, $\beta_2 = 0.33$ and $\beta_3 = 0.01$ gives that $KMv \approx 48.4 \frac{a^4b^2}{\epsilon^3} \frac{\log \frac{ac}{\epsilon} + 5.03}{(-\log \alpha)}$. Finally, we can give an explicit schedule for $K$, $M$, and $v$, and bound the total amount of work that needs to be performed.

**Theorem 5.** *If $D \geq \max\left(\|\theta_0 - \theta^*\|_2, \frac{4R_2}{L}\log\frac{1}{\delta}\right)$, then for all $\epsilon$ there is a setting of $K, M, v$ such that $f(\frac{1}{K}\sum_{k=1}^{K}\theta_k) - f(\theta^*) \leq \epsilon_f$ with probability $1 - \delta$ and*

$$KMv \leq \frac{32LR_2^2D^4}{\beta_1^4\beta_2^2\epsilon_f^3(1-\alpha)} \log \frac{4DR_2C}{\beta_1\beta_3\epsilon_f}.$$

*[Proof sketch] This follows from setting $K$, $M$, and $v$ as in Theorem 4 with $a = L\|\theta_0 - \theta^*\|_2/(4R_2) + \log\frac{1}{\delta}$, $b = 1$, $c = C$, and $\epsilon = \epsilon_f L/(8R_2^2)$.*

### 4.3 Strongly Convex Convergence

This section gives the main result for convergence that is true only in the regularized case where $\lambda > 0$. Again, the main difficulty in this proof is showing that the sum of the errors of estimated gradients at each iteration is small. This is done by using a concentration inequality to show that the error of each estimated gradient is small, and then applying a union bound to show that the sum is small. The main result is as follows.

**Theorem 6.** *When the regularization constant obeys $\lambda > 0$, with probability at least $1 - \delta$ Algorithm 1 will satisfy*

$$\|\theta_K - \theta^*\|_2 \leq (1 - \frac{\lambda}{L})^K \|\theta_0 - \theta^*\|_2 + \frac{L}{\lambda}\left(\sqrt{\frac{R_2}{2M}}\left(1 + \sqrt{2\log\frac{K}{\delta}}\right) + 2R_2C\alpha^v\right).$$

*Proof sketch.* When $\lambda = 0$, $f$ is convex (as in Theorem 2) and so is strongly convex when $\lambda > 0$. The basic proof technique here is to decompose the error in a particular step as $\|e_{k+1}\|_2 \leq \|\frac{1}{M}\sum_{i=1}^{M} t(x_i^k) - \mathbb{E}_{q_k}[t(X)]\|_2 + \|\mathbb{E}_{q_k}[t(X)] - \mathbb{E}_{p_{\theta_k}}[t(X)]\|_2$. A multidimensional variant of Hoeffding's inequality can bound the first term, with probability $1 - \delta'$ by $R_2(1 + \sqrt{2\log\frac{1}{\delta}})/\sqrt{M}$, while our assumption on mixing speed can bound the second term by $2R_2C\alpha^v$. Applying this to all iterations using $\delta' = \delta/K$ gives that all errors are simultaneously bounded as before. This can then be used in another result due to Schmidt et al. [26] on the convergence of gradient descent with errors in estimated gradients in the strongly convex case. $\square$

A similar proof strategy could be used for the convex case where, rather than directly bounding the sum of the norm of errors of all steps using the Efron-Stein inequality and Bernstein's bound, one could simply bound the error of each step using a multidimensional Hoeffding-type inequality, and then apply this with probability $\delta/K$ to each step. This yields a slightly weaker result than that shown in Theorem 2. The reason for applying a uniform bound on the errors in gradients here is that Schmidt et al.'s bound [26] on the convergence of proximal gradient descent on strongly convex functions depends not just on the sum of the norms of gradient errors, but a non-uniform weighted variant of these.

Again, we consider how to set parameters to guarantee that $\theta_K$ is not too far from $\theta^*$ with a minimum amount of work. Firstly, we show a lower-bound.

**Theorem 7.** *Suppose $a, b, c > 0$. Then for any $K, M, v$ such that $\gamma^K a + \frac{b}{\sqrt{M}}\sqrt{\log(K/\delta)} + c\alpha^v \leq \epsilon$. it must be the case that*

$$KMv \geq \frac{b^2}{\epsilon^2}\frac{\log\frac{a}{\epsilon}\log\frac{c}{\epsilon}}{(-\log\gamma)(-\log\alpha)}\log\left(\frac{\log\frac{a}{\epsilon}}{\delta(-\log\gamma)}\right).$$

*[Proof sketch] This is established by noticing that $\gamma^K a$, $\frac{b}{\sqrt{M}}\sqrt{\log\frac{K}{\delta}}$, and $c\alpha^v$ must each be less than $\epsilon$, giving lower bounds on $K$, $M$, and $v$.*

Next, we can give an explicit schedule that is not too far off from this lower-bound.

**Theorem 8.** *Suppose that $a, b, c, \alpha > 0$. If $\beta_1 + \beta_2 + \beta_3 = 1$, $\beta_i > 0$, then setting $K = \log(\frac{a}{\beta_1\epsilon})/(-\log\gamma)$, $M = \frac{b^2}{\epsilon^2\beta_2^2}\left(1 + \sqrt{2\log(K/\delta)}\right)^2$ and $v = \log\left(\frac{c}{\beta_3\epsilon}\right)/(-\log\alpha)$ is sufficient to guarantee that $\gamma^K a + \frac{b}{\sqrt{M}}(1 + \sqrt{2\log(K/\delta)}) + c\alpha^v \leq \epsilon$ with a total work of at most*

$$KMV \leq \frac{b^2}{\epsilon^2\beta_2^2}\frac{\log\left(\frac{a}{\beta_1\epsilon}\right)\log\left(\frac{c}{\beta_3\epsilon}\right)}{(-\log\gamma)(-\log\alpha)}\left(1 + \sqrt{2\log\frac{\log(\frac{a}{\beta_1\epsilon})}{\delta(-\log\gamma)}}\right)^2.$$

For example, if you choose $\beta_2 = 1/\sqrt{2}$ and $\beta_1 = \beta_3 = (1 - 1/\sqrt{2})/2 \approx 0.1464$, then this varies from the lower-bound in Theorem 7 by a factor of two, and a multiplicative factor of $1/\beta_3 \approx 6.84$ inside the logarithmic terms.

**Corollary 9.** *If we choose $K \geq \frac{L}{\lambda}\log\left(\frac{\|\theta_0 - \theta\|_2}{\beta_1\epsilon}\right)$, $M \geq \frac{L^2 R_2}{2\epsilon^2\beta_2^2\lambda^2}\left(1 + \sqrt{2\log(K/\delta)}\right)^2$, and $v \geq \frac{1}{1-\alpha}\log\left(2LR_2C/(\beta_3\epsilon\lambda)\right)$, then $\|\theta_K - \theta^*\|_2 \leq \epsilon_\theta$ with probability at least $1 - \delta$, and the total amount of work is bounded by*

$$KMv \leq \frac{L^3 R_2}{2\epsilon_\theta^2\beta_2^2\lambda^3(1-\alpha)}\log\left(\frac{\|\theta_0 - \theta\|_2}{\beta_1\epsilon_\theta}\right)\left(1 + \sqrt{2\log\left(\frac{L}{\lambda\delta}\log\left(\frac{\|\theta_0 - \theta\|_2}{\beta_1\epsilon_\theta}\right)\right)}\right)^2.$$

## 5 Discussion

An important detail in the previous results is that the convex analysis gives convergence in terms of the regularized log-likelihood, while the strongly-convex analysis gives convergence in terms of the parameter distance. If we drop logarithmic factors, the amount of work necessary for $\epsilon_f$ - optimality in the log-likelihood using the convex algorithm is of the order $1/\epsilon_f^3$, while the amount of work necessary for $\epsilon_\theta$ - optimality using the strongly convex analysis is of the order $1/\epsilon_\theta^2$. Though these quantities are not directly comparable, the standard bounds on sub-optimality for $\lambda$-strongly convex functions with $L$-Lipschitz gradients are that $\lambda\epsilon_\theta^2/2 \leq \epsilon_f \leq L\epsilon_\theta^2/2$. Thus, roughly speaking, when regularized for the strongly-convex analysis shows that $\epsilon_f$ optimality in the log-likelihood can be achieved with an amount of work only linear in $1/\epsilon_f$.

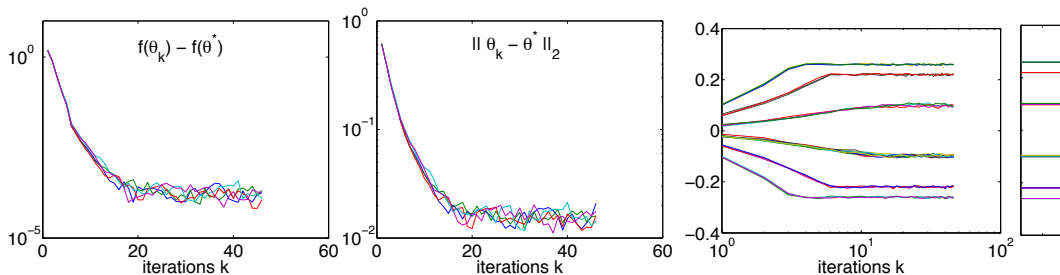

Figure 2: Ising Model Example. Left: The difference of the current test log-likelihood from the optimal log-likelihood on 5 random runs. Center: The distance of the current estimated parameters from the optimal parameters on 5 random runs. Right: The current estimated parameters on one run, as compared to the optimal parameters (far right).

## 6 Example

While this paper claims no significant practical contribution, it is useful to visualize an example. Take an Ising model $p(x) \propto \exp(\sum_{(i,j)\in\text{Pairs}} \theta_{ij} x_i x_j)$ for $x_i \in \{-1,1\}$ on a $4 \times 4$ grid with 5 random vectors as training data. The sufficient statistics are $t(x) = \{x_i x_j | (i,j) \in \text{Pairs}\}$, and with 24 pairs, $\|t(x)\|_2 \leq R_2 = \sqrt{24}$. For a fast-mixing set, constrain $|\theta_{ij}| \leq .2$ for all pairs. Since the maximum degree is 4, $\tau(\epsilon) \leq \lceil \frac{N \log(N/\epsilon)}{1-4\tanh(.2)} \rceil$. Fix $\lambda = 1$, $\epsilon_\theta = 2$ and $\delta = 0.1$. Though the theory above suggests the Lipschitz constant $L = 4R_2^2 + \lambda = 97$, a lower value of $L = 10$ is used, which converged faster in practice (with exact or approximate gradients). Now, one can derive that $\|\theta_0 - \theta^*\|_2 \leq D = \sqrt{24 \times (2 \times .2)^2}$, $C = \log(16)$ and $\alpha = \exp(-(1 - 4\tanh .2)/16)$. Applying Corollary 9 with $\beta_1 = .01$, $\beta_2 = .9$ and $\beta_3 = .1$ gives $K = 46$, $M = 1533$ and $v = 561$. Fig. 2 shows the results. In practice, the algorithm finds a solution tighter than the specified $\epsilon_\theta$, indicating a degree of conservatism in the theoretical bound.

## 7 Conclusions

This section discusses some weaknesses of the above analysis, and possible directions for future work. Analyzing complexity in terms of the total sampling effort ignores the complexity of projection itself. Since projection only needs to be done $K$ times, this time will often be very small in comparison to sampling time. (This is certainly true in the above example.) However, this might not be the case if the projection algorithm scales super-linearly in the size of the model.

Another issue to consider is how the samples are initialized. As far as the proof of correctness goes, the initial distribution $r$ is arbitrary. In the above example, a simple uniform distribution was used. However, one might use the empirical distribution of the training data, which is equivalent to contrastive divergence [5]. It is reasonable to think that this will tend to reduce the mixing time when the $p_\theta$ is close to the model generating the data. However, the number of Markov chain transitions $v$ prescribed above is larger than typically used with contrastive divergence, and Algorithm 1 does not reduce the step size over time. While it is common to regularize to encourage fast mixing with contrastive divergence [14, Section 10], this is typically done with simple heuristic penalties. Further, contrastive divergence is often used with hidden variables. Still, this provides a bound for how closely a variant of contrastive divergence could approximate the maximum likelihood solution.

The above analysis does not encompass the common strategy for maximum likelihood learning where one maintains a "pool" of samples between iterations, and initializes one Markov chain at each iteration from each element of the pool. The idea is that if the samples at the previous iteration were close to $p_{k-1}$ and $p_{k-1}$ is close to $p_k$, then this provides an initialization close to the current solution. However, the proof technique used here is based on the assumption that the samples $x_i^k$ at each iteration are independent, and so cannot be applied to this strategy.

### Acknowledgements
Thanks to Ivona Bezáková, Aaron Defazio, Nishant Mehta, Aditya Menon, Cheng Soon Ong and Christfried Webers. NICTA is funded by the Australian Government through the Dept. of Communications and the Australian Research Council through the ICT Centre of Excellence Program.

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
