[Supplementary Material]

# Appendix

## 8 Background

### 8.1 Optimization

The main results in this paper rely strongly on the work of Schmidt et al. [26] on the convergence of proximal gradient methods with errors in estimated gradients. The first result used is the following theorem for the convergence of gradient descent on convex functions with errors in the estimated gradients.

**Theorem 10.** *(Special case of [26, Proposition 1]) Suppose that a function $f$ is convex with an $L$-Lipshitz gradient (meaning $\|f'(\phi) - f'(\theta)\|_2 \leq L\|\phi - \theta\|_2$). If $\Theta$ is a closed convex set and one iterates*

$$\theta_k \leftarrow \Pi_\Theta \left[ \theta_{k-1} - \frac{1}{L} \left( f'(\theta_{k-1}) + e_k \right) \right],$$

*then, defining $\theta^* \in \arg\min_{\theta \in \Theta} f(\theta)$, for all $K \geq 1$, we have, for $A_K := \sum_{k=1}^{K} \frac{\|e_k\|}{L}$, that*

$$f\left( \frac{1}{K} \sum_{k=1}^{K} \theta_k \right) - f(\theta^*) \quad \leq \quad \frac{L}{2K} \left( \|\theta_0 - \theta^*\|_2 + 2A_K \right)^2.$$

This section will show that this is indeed a special case of [26]. To start with, we simply restate exactly the previous result [26, Proposition 1], with only trivial changes in notation.

**Theorem 11.** *Assume that:*

- *$f$ is convex and has $L$-Lipschitz continuous gradient*

- *$h$ is a lower semi-continuous proper convex function.*

- *The function $r = f + h$ attains it's minimum at a certain $\theta^* \in \mathbb{R}^n$.*

- *$\theta_k$ is an $\epsilon_k$-optimal solution, i.e. that*

$$\frac{L}{2}\|\theta_k - y\|^2 + h(\theta_k) \leq \epsilon_k + \min_{\theta \in \mathbb{R}^n} \frac{L}{2}\|\theta - y\|^2 + h(\theta)$$

  *where*

$$y = \theta_{k-1} - \frac{1}{L} \left( f'(\theta_{k-1}) + e_k \right).$$

*Then, for all $K \geq 1$, one has that*

$$r\left( \frac{1}{K} \sum_{k=1}^{K} \theta_k \right) - r(\theta^*) \leq \frac{L}{2K} \left( \|\theta_0 - \theta^*\| + 2A_K + \sqrt{2B_K} \right)^2$$

*with*

$$A_K = \sum_{k=1}^{K} \left( \frac{\|e_k\|}{L} + \sqrt{\frac{2\epsilon_k}{L}} \right), \quad B_K = \sum_{k=1}^{K} \frac{\epsilon_k}{K}.$$

The first theorem follows from this one by setting $h$ to be the indicator function for the set $\Theta$, i.e.

$$h(\theta) = \begin{cases} 0 & \theta \in \Theta \\ \infty & \theta \notin \Theta \end{cases}$$

and assuming that $\epsilon_k = 0$. By the convexity of $\Theta$, $h$ will be a lower semi-continuous proper convex function. Further, from the fact that $\Theta$ is closed, $r$ will attain its minimum. Now, we verify that this

results in the theorem statement at the start of this section. $\theta_k$ takes the form

$$
\begin{aligned}
\theta_k &= \arg\min_{\theta\in\mathbb{R}^n} \frac{L}{2}\|\theta - y\|^2 + h(\theta) \\
&= \arg\min_{\theta\in\Theta} \|\theta - y\| \\
&= \arg\min_{\theta\in\Theta} \|\theta - \theta_{k-1} + \frac{1}{L}\left(f'(\theta_{k-1}) + e_k\right)\| \\
&= \Pi_\Theta \left[\theta_{k-1} - \frac{1}{L}\left(f'(\theta_{k-1}) + e_k\right)\right].
\end{aligned}
$$

We will also use the following result for strongly-convex optimzation. The special case follows from the same construction used above.

Next, consider the following result on optimization of strongly convex functions, which follows from [26] by a very similar argument.

**Theorem 12.** *(Special case of [26, Proposition 3]) Suppose that a function $f$ is $\lambda$-strongly convex with an L-Lipshitz gradient (meaning $\|f'(\phi) - f'(\theta)\|_2 \leq L\|\phi - \theta\|_2$). If $\Theta$ is a closed convex set and one iterates*

$$
\theta_k \leftarrow \Pi_\Theta \left[\theta_{k-1} - \frac{1}{L}\left(f'(\theta_{k-1}) + e_k\right)\right],
$$

*Then, defining $\theta^* = \arg\min_{\theta\in\Theta} f(\theta)$, for all $K \geq 1$, we have, for $\bar{A}_k = \sum_{k=1}^{K}(1 - \frac{\lambda}{L})^{-k}\frac{\|e_k\|}{L}$ that*

$$
\|\theta_K - \theta^*\|_2 \leq (1 - \frac{\lambda}{L})^K \left(\|\theta_0 - \theta^*\|_2 + \bar{A}_k\right)
$$

**Corollary 13.** *Under the same conditions, if $\|e_k\| \leq r$ for all k, then*

$$
\|\theta_K - \theta^*\|_2 \leq (1 - \frac{\lambda}{L})^K \|\theta_0 - \theta^*\|_2 + \frac{rL}{\lambda}
$$

*Proof.* Using the fact that $\sum_{k=1}^{K} a^{-k} = a^{-K}\sum_{k=0}^{K-1} a^k \leq a^{-K}\sum_{k=0}^{\infty} a^k = \frac{a^{-K}}{1-a}$, we get that

$$
\bar{A}_K \leq r\sum_{k=1}^{K}(1 - \frac{\lambda}{L})^{-k} \leq r\frac{L}{\lambda}(1 - \frac{\lambda}{L})^{-K},
$$

and therefore that

$$
\|\theta_K - \theta^*\|_2 \leq (1 - \frac{\lambda}{L})^K \left(\|\theta_0 - \theta^*\|_2 + r\frac{L}{\lambda}(1 - \frac{\lambda}{L})^{-K}\right).
$$

$\square$

## 8.2 Concentration Results

Three concentration inequalities, are stated here for reference. The first is Bernstein's inequality.

**Theorem 14.** *(Bernstein's inequality) Suppose $Z_1, ..., Z_K$ are independent with mean 0, that $|Z_k| \leq c$ and that $\sigma_i^2 = \mathbb{V}[Z_i]$. Then, if we define $\sigma^2 = \frac{1}{K}\sum_{k=1}^{K}\sigma_k^2$,*

$$
\mathbb{P}\left[\frac{1}{K}\sum_{k=1}^{K} Z_k > \epsilon\right] \leq \exp\left(-\frac{K\epsilon^2}{2\sigma^2 + 2c\epsilon/3}\right).
$$

The second is the following Hoeffding-type bound to control the difference between the expected value of $t(X)$ and the estimated value using $M$ samples.

**Theorem 15.** *If $X_1, ..., X_M$ are independent variables with mean $\mu$, and $\|X_i - \mu\| \leq c$, then for all $\epsilon \geq 0$, with probability at least $1 - \delta$,*

$$
\|\bar{X} - \mu\| \leq \sqrt{\frac{c}{4M}}\left(1 + \sqrt{2\log\frac{1}{\delta}}\right).
$$

*Proof.* Boucheron et al. [2013, Ex. 6.3] show that, under the same conditions as stated, for all $s \geq \sqrt{v}$,

$$\mathbb{P}\left[\|\bar{X} - \mu\| > \frac{s}{M}\right] \leq \exp\left(-\frac{(s - \sqrt{v})^2}{2v}\right),$$

where $v = \frac{cM}{4}$. We will fix $\delta$, and solve for the appropriate $s$. If we set $\delta = \exp(-\frac{(s-\sqrt{v})^2}{2v})$, then we have that $s = \sqrt{2v \log \frac{1}{\delta}} + \sqrt{v}$, meaning that, with probability at least $1 - \delta$,

$$\|\bar{X} - \mu\| \quad \leq \quad \frac{1}{M}\left(\sqrt{2\frac{cM}{4}\log\frac{1}{\delta}} + \sqrt{\frac{cM}{4}}\right),$$

which is equivalent to the result with a small amount of manipulation. $\qquad\square$

The third is the Efron-Stein inequality [4, Theorem 3.1].

**Theorem 16.** *If $X = (X_1, ..., X_m)$ is a vector of independent random variables and $f(X)$ is a square-integrable function, then*

$$\mathbb{V}[f(X)] \leq \frac{1}{2}\sum_{i=1}^{M}\mathbb{E}\left[\left((f(X) - f(X^{(i)}))\right)^2\right],$$

*where $X^{(i)}$ is $X$ with $X_i$ independently re-drawn, i.e.*

$$X^{(i)} = (X_1, ..., X_{i-1}, X'_{i'}, X_{i+1}, ..., X_m).$$

# 9   Preliminary Results

A result that we will use several times below is that, for $0 < \alpha < 1$, $-\frac{1}{\log(\alpha)} \leq \frac{1}{1-\alpha}$. This bound is tight in the limit that $\alpha \to 1$.

**Lemma 17.** *The difference of two estimated mean vectors is bounded by*

$$\|\mathbb{E}_q[t(X)] - \mathbb{E}_p[t(X)]\|_2 \leq 2R_2\|q - p\|_{TV}.$$

*Proof.* Let the distribution functions of $p$ and $q$ be $P$ and $Q$, respectively. Then, we have that

$$\|\mathbb{E}_p[t(X)] - \mathbb{E}_q[t(X)]\|_2 = \left\|\int_x t(x)\,(dP(x) - dQ(x))\right\|_2$$
$$\leq \int_x |dP(x) - dQ(x)| \cdot \|t(x)\|_2.$$

Using the definition of total-variation distance, and the bound that $\|t(x)\|_2 \leq R_2$ gives the result. $\qquad\square$

**Lemma 18.** *If $1/a + 1/b = 1$, then the difference of two log-partition functions is bounded by*

$$|A(\theta) - A(\phi)| \leq R_a\|\theta - \phi\|_b.$$

*Proof.* By the Lagrange remainder theorem, there must exist some $\gamma$ on the line segment between $\theta$ and $\phi$ such that $A(\phi) = A(\theta) + (\phi - \theta)^T \nabla_\gamma A(\gamma)$. Thus, applying Hölder's inequality, we have that

$$|A(\phi) - A(\theta)| = |(\phi - \theta)^T \nabla_\gamma A(\gamma)| \leq \|\phi - \theta\|_b \cdot \|\nabla_\gamma A(\gamma)\|_a.$$

The result follows from the fact that $\|\nabla_\gamma A(\gamma)\|_a = \|\mathbb{E}_{p_\gamma} t(X)\|_a \leq R_a$. $\qquad\square$

Next, we observe that the total variation distance between $p_\theta$ and $p_\phi$ is bounded by the distance between $\theta$ and $\phi$.

**Theorem 19.** *If $1/a + 1/b = 1$, then the difference of distributions is bounded by*

$$\|p_\theta - p_\phi\|_{TV} \leq 2R_a\|\theta - \phi\|_b.$$

*Proof.* If we assume that $p_\theta$ is a density, we can decompose the total-variation distance as

$$\|p_\theta - p_\phi\|_{TV}$$
$$= \frac{1}{2} \int_x p_\theta(x) |1 - \frac{p_\phi(x)}{p_\theta(x)}|$$
$$= \frac{1}{2} \int_x p_\theta(x) |1 - \exp\left((\phi - \theta) \cdot t(x) - A(\phi) + A(\theta)\right)|$$
$$\leq \frac{1}{2} \int_x p_\theta(x) |1 - \exp|(\phi - \theta) \cdot t(x) - A(\phi) + A(\theta)||.$$

If $p_\theta$ is a distribution, the analogous expression is true, replacing the integral over $x$ with a sum.

We can upper-bound the quantity inside $\exp$ by applying Hölder's inequality and the previous Lemma as

$$|(\phi - \theta) \cdot t(x) - A(\phi) + A(\theta)|$$
$$\leq |(\phi - \theta) \cdot t(x)| + |A(\phi) - A(\theta))|$$
$$\leq 2R_a\|\theta - \phi\|_b.$$

From which we have that

$$\|p_\theta - p_\phi\|_{TV} \leq \frac{1}{2} |1 - \exp\left(2R_a\|\theta - \phi\|_b\right)|.$$

If $2R_a\|\theta - \phi\|_b > 1$, the theorem is obviously true, since $\| \cdot \|_{TV} \leq 1$. Suppose instead that that $2R_a\|\theta - \phi\|_b \leq 1$. If $0 \leq c \leq 1$, then $\frac{1}{2}|1 - \exp(c)| \leq c\frac{e-1}{2}$. Applying this with $c = 2R_a\|\theta - \phi\|_b$ gives that $\|p_\theta - p_\phi\|_{TV} \leq (e-1)R_2\|\theta - \phi\|_b$. The result follows from the fact that $2 > (e-1)$. $\square$

## 10  Lipschitz Continuity

This section shows that the ridge-regularized empirical log-likelihood does indeed have a Lipschitz continuous gradient.

**Theorem 20.** *The regularized log-likelihood function is L-Lipschitz with $L = 4R_2^2 + \lambda$, i.e.*

$$\|f'(\theta) - f'(\phi)\|_2 \leq (4R_2^2 + \lambda)\|\theta - \phi\|_2.$$

*Proof.* We start by the definition of the gradient, with

$$\|f'(\theta) - f'(\phi)\|_2 = \left\|\left(\frac{dA}{d\theta} - \bar{t} + \lambda\theta\right) - \left(\frac{dA}{d\phi} - \bar{t} + \lambda\phi\right)\right\|_2$$
$$= \|\frac{dA}{d\theta} - \frac{dA}{d\phi} + \lambda(\theta - \phi)\|_2.$$
$$\leq \|\frac{dA}{d\theta} - \frac{dA}{d\phi}\|_2 + \lambda\|\theta - \phi\|_2.$$

Now, looking at the first two terms, we can apply Lemma 17 to get that

$$\left\|\frac{dA}{d\theta} - \frac{dA}{d\phi}\right\|_2 = \left\|\mathbb{E}_{p_\theta}[t(X)] - \mathbb{E}_{p_\phi}[t(X)]\right\|_2$$
$$\leq 2R_2\|p_\theta - p_\phi\|_{TV}.$$

Observing by Theorem 19 that $\|p_\theta - p_\phi\|_{TV} \leq 2R_2\|\theta - \phi\|_2$ gives that

$$\|f'(\theta) - f'(\phi)\|_2 \leq 4R_2^2\|\theta - \phi\|_2 + \lambda\|\theta - \phi\|_2$$

$\square$

## 11 Convex Convergence

This section gives the main result for convergence this is true both in the regularized case where $\lambda > 0$ and the unregularized case where $\lambda = 0$. The main difficulty in this proof is showing that the sum of the norms of the errors of estimated gradients is small.

**Theorem 21.** *Assuming that $X_1, ..., X_M$ are independent and identically distributed with mean $\mu$ and that $\|X_m\|_2 \leq R_2$, then*

$$\mathbb{E}\left[\|\frac{1}{M}\sum_{m=1}^{M} X_m - \mu\|_2\right] \leq \frac{2R_2}{\sqrt{M}}$$

*Proof.* Using that $\mathbb{E}\left[Z^2\right] = \mathbb{V}\left[Z\right] + \mathbb{E}\left[Z\right]^2$ and the fact that the variance is non-negative (Or simply Jensen's inequality), we have

$$
\begin{aligned}
\mathbb{E}\left[\|\frac{1}{M}\sum_{m=1}^{M} X_m - \mu\|_2\right]^2 &\leq \mathbb{E}\left[\|\frac{1}{M}\sum_{m=1}^{M} X_m - \mu\|_2^2\right] \\
&= \frac{1}{M}\mathbb{E}\left[\|X_m - \mu\|_2^2\right] \\
&\leq \frac{1}{M}(2R_2)^2 \\
&= \frac{4R_2^2}{M}.
\end{aligned}
$$

Taking the square-root gives the result. $\qquad\square$

**Theorem 22.** *Assuming that $X_1, ..., X_M$ are iid with mean $\mu$ and that $\|X_m\| \leq R_2$, then*

$$\mathbb{V}\left[\|\frac{1}{M}\sum_{m=1}^{M} X_m - \mu\|\right] \leq \frac{2R_2^2}{M}.$$

*Proof.*

$$
\begin{aligned}
\mathbb{V}\left[\|\frac{1}{M}\sum_{m=1}^{M} X_m - \mu\|\right] &= \mathbb{V}\left[\|\frac{1}{M}\sum_{m=1}^{M} (X_m - \mu)\|\right] \\
&= \frac{1}{M^2}\mathbb{V}\left[\|\sum_{m=1}^{M} (X_m - \mu)\|\right]
\end{aligned}
$$

Now, the Efron-Stein inequality tells us that

$$\mathbb{V}[f(X_1, ..., X_m)] \leq \frac{1}{2}\sum_{m'=1}^{M}\mathbb{E}\left[\left((f(X) - f(X^{(m')})\right)^2\right]$$

where $X^{(m')}$ is $X$ with $X_{m'}$ independently re-drawn. Now, we identify $f(X_1, ..., X_m) = \|\sum_{m=1}^{M}(X_m - \mu)\|$ to obtain that

$$\mathbb{V}\left[\|\sum_{m=1}^{M} (X_m - \mu)\|\right] \leq \frac{1}{2}\sum_{m'=1}^{M}\mathbb{E}\left[\left(\|\sum_{m=1}^{M} (X_m - \mu)\| - \|\sum_{m=1}^{M} (X_m^{(m')} - \mu)\|\right)^2\right].$$

Further, since we know that

$$\sum_{m=1}^{M} (X_m^{(m')} - \mu) = \sum_{m=1}^{M} (X_m - \mu) + X_{m'}^{(m')} - X_{m'},$$

we can apply that that $(\|a + b\| - \|a\|)^2 \leq \|b\|^2$ to obtain that

$$\left( \| \sum_{m=1}^{M} (X_m - \mu)\| - \| \sum_{m=1}^{M} (X_m^{(m')} - \mu)\| \right)^2 = \|X_{m'}^{(m')} - X_{m'}\|^2,$$

and so

$$\mathbb{V}\left[ \| \sum_{m=1}^{M} (X_m - \mu)\| \right] \leq \frac{1}{2} \sum_{m'=1}^{M} \mathbb{E}\left[ \|X_{m'}^{(m')} - X_{m'}\|^2 \right].$$

And, since we assume that $\|X_m\| \leq R_2$, $\|X_{m'}^{(m')} - X_{m'}\| \leq 2R_2$, which leads to

$$\mathbb{V}\left[ \| \sum_{m=1}^{M} (X_m - \mu)\| \right] \leq 2MR_2^2,$$

from which it follows that

$$\mathbb{V}\left[ \| \frac{1}{M} \sum_{m=1}^{M} X_m - \mu\| \right] \leq \frac{2R_2^2}{M}.$$

$\square$

**Theorem 23.** *With probability at least $1 - \delta$,*

$$\sum_{k=1}^{K} \| \frac{1}{M} \sum_{i=1}^{M} t(x_i^k) - \mathbb{E}_{q_k}[t(X)]\|_2 \leq K\epsilon(\delta) + \frac{2R_2 K}{\sqrt{M}},$$

*where $\epsilon(\delta)$ is the solution to*

$$\delta = \exp\left( -\frac{K\epsilon^2}{4R_2^2/M + 4R_2\epsilon/3} \right). \tag{5}$$

*Proof.* Let $d_k = \frac{1}{M} \sum_{i=1}^{M} t(x_i^k) - \mathbb{E}_{q_k}[t(X)]$. Applying Bernstein's inequality immediately gives us that

$$\mathbb{P}\left[ \frac{1}{K} \sum_{k=1}^{K} (\|d_k\|_2 - \mathbb{E}\|d_k\|_2) > \epsilon \right] \leq \exp\left( -\frac{K\epsilon^2}{2\sigma^2 + 2c\epsilon/3} \right).$$

Here, we can bound $\sigma^2$ by

$$\sigma^2 = \frac{1}{K} \sum_{k=1}^{K} \sigma_k^2 = \frac{1}{K} \sum_{k=1}^{K} \mathbb{V}\left[ \|d_k\|_2 - \mathbb{E}\|d_k\|_2 \right] = \frac{1}{K} \sum_{k=1}^{K} \mathbb{V}\left[ \|d_k\|_2 \right] \leq \frac{2R_2^2}{M},$$

where the final inequality follows from Theorem 22. We also know that $\|d_k\| \leq 2R_2 = c$, from which we get that

$$\mathbb{P}\left[ \frac{1}{K} \sum_{k=1}^{K} \|d_k\|_2 - \mathbb{E}[\|d_k\|_2] > \epsilon \right] \leq \exp\left( -\frac{K\epsilon^2}{4R_2^2/M + 4R_2\epsilon/3} \right).$$

So we have that, with probability $1 - \delta$

$$\frac{1}{K} \sum_{k=1}^{K} \|d_k\|_2 - \mathbb{E}[\|d_k\|_2] \leq \epsilon(\delta)$$

$$\frac{1}{K} \sum_{k=1}^{K} \|d_k\|_2 \leq \epsilon(\delta) + \mathbb{E}[\|d_k\|_2]$$

$$\leq \epsilon(\delta) + \frac{2R_2}{\sqrt{M}},$$

where the final inequality follows from Theorem 21.

$\square$

**Corollary 24.** *If* $M \geq 3K/\log(\frac{1}{\delta})$, *then with probability at least* $1 - \delta$,

$$\sum_{k=1}^{K} \|\frac{1}{M} \sum_{i=1}^{M} t(x_i^k) - \mathbb{E}_{q_k}[t(X)]\|_2 \leq 2R_2 \left( \frac{K}{\sqrt{M}} + \log \frac{1}{\delta} \right).$$

*Proof.* Solving Equation 5 for $\epsilon$ yields that

$$\epsilon(\delta) = \frac{2R_2}{3K} \left( \log \frac{1}{\delta} + \sqrt{\left( \log \frac{1}{\delta} \right)^2 + \frac{9K \log \frac{1}{\delta}}{M}} \right).$$

Now, suppose that $\frac{3K}{M} \leq \log \frac{1}{\delta}$, as assumed here. Then,

$$
\begin{aligned}
\epsilon(\delta) &\leq \frac{2R_2}{3K} \left( \log \frac{1}{\delta} + \sqrt{\left( \log \frac{1}{\delta} \right)^2 + 3(\log \frac{1}{\delta})^2} \right) \\
&\leq \frac{2R_2}{3K} \left( \log \frac{1}{\delta} + 2\log(\frac{1}{\delta}) \right) \\
&= \frac{2R_2}{K} \log \frac{1}{\delta}.
\end{aligned}
$$

Substituting this bound into the result of Theorem 23 gives the result. $\square$

Now, we can prove the main result.

**Theorem 25.** *With probability at least* $1 - \delta$, *at long as* $M \geq 3K/\log(\frac{1}{\delta})$,

$$f\left( \frac{1}{K} \sum_{k=1}^{K} \theta_k \right) - f(\theta^*) \leq \frac{8R_2^2}{KL} \left( \frac{L\|\theta_0 - \theta^*\|_2}{4R_2} + \log \frac{1}{\delta} + \frac{K}{\sqrt{M}} + KC\alpha^v \right)^2.$$

*Proof.* Applying Theorem 10 gives that

$$f\left( \frac{1}{K} \sum_{k=1}^{K} \theta_k \right) - f(\theta^*) \leq \frac{L}{2K} \left( \|\theta_0 - \theta^*\|_2 + 2A_K \right)^2,$$

for $A_K = \frac{1}{L} \sum_{k=1}^{K} \|e_k\|$, where

$$
\begin{aligned}
e_k &= \frac{1}{M} \sum_{i=1}^{M} t(x_i^{k-1}) - \bar{t} + \lambda \theta_{k-1} - f'(\theta_{k-1}) \\
&= \frac{1}{M} \sum_{i=1}^{M} t(x_i^{k-1}) - \mathbb{E}_{p_{k-1}}[t(X)].
\end{aligned}
$$

Now, we know that

$$\sum_{k=1}^{K} \|e_k\| \leq \sum_{k=1}^{K} \|\frac{1}{M} \sum_{i=1}^{M} t(x_i^{k-1}) - \mathbb{E}_{q_{k-1}}[t(X)]\|_2 + \sum_{k=1}^{K} \|\mathbb{E}_{q_{k-1}}[t(X)] - \mathbb{E}_{p_{k-1}}[t(X)]\|_2.$$

We have by Lemma 17 and the assumption of mixing speed that

$$\|\mathbb{E}_{q_{k-1}}[t(X)] - \mathbb{E}_{p_{k-1}}[t(X)]\|_2 \leq 2R_2 \|q_{k-1} - p_{k-1}\|_{TV} \leq 2R_2 C\alpha^v.$$

Meanwhile, the previous Corollary tells us that, with probability $1 - \delta$,

$$\sum_{k=1}^{K} \|\frac{1}{M} \sum_{i=1}^{M} t(x_i^{k-1}) - \mathbb{E}_{q_{k-1}}[t(X)]\|_2 \leq 2R_2 \left( \frac{K}{\sqrt{M}} + \log \frac{1}{\delta} \right)$$

Thus, we have that

$$
\begin{aligned}
f\left(\frac{1}{K}\sum_{k=1}^{K}\theta_k\right) - f(\theta^*) &\leq \frac{L}{2K}\left(\|\theta_0 - \theta^*\|_2 + \frac{2}{L}\left(2R_2\left(\frac{K}{\sqrt{M}} + \log\frac{1}{\delta}\right) + 2R_2KC\alpha^v\right)\right)^2 \\
&= \frac{L}{2K}\left(\|\theta_0 - \theta^*\|_2 + \frac{4R_2}{L}\left(\frac{K}{\sqrt{M}} + \log\frac{1}{\delta} + KC\alpha^v\right)\right)^2 \\
&= \frac{8R_2^2}{KL}\left(\frac{L\|\theta_0 - \theta^*\|_2}{4R_2} + \log\frac{1}{\delta} + \frac{K}{\sqrt{M}} + KC\alpha^v\right)^2.
\end{aligned}
$$

$\square$

Now, what we really want to do is guarantee that $f\left(\frac{1}{K}\sum_{k=1}^{K}\theta_k\right) - f(\theta^*) \leq \epsilon$, while ensuring the the total work $MKv$ is not too large. Our analysis will use the following theorem.

**Theorem 26.** *Suppose that $a, b, c, \alpha > 0$. If $\beta_1 + \beta_2 + \beta_3 = 1$, $\beta_1, \beta_2, \beta_3 > 0$, then setting*

$$
K = \frac{a^2}{\beta_1^2\epsilon}, \quad M = (\frac{ab}{\beta_1\beta_2\epsilon})^2, \quad v = \frac{\log\frac{ac}{\beta_1\beta_3\epsilon}}{(-\log\alpha)}
$$

*is sufficient to guarantee that $\frac{1}{K}\left(a + b\frac{K}{\sqrt{M}} + Kc\alpha^v\right)^2 \leq \epsilon$ with a total work of*

$$
KMv = \frac{1}{\beta_1^4\beta_2^2}\frac{a^4b^2}{\epsilon^3}\frac{\log\frac{ac}{\beta_1\beta_3\epsilon}}{(-\log\alpha)}.
$$

*Proof.* Firstly, we should verify the $\epsilon$ bound. We have that

$$
\begin{aligned}
a + b\frac{K}{\sqrt{M}} + Kc\alpha^v &= a + b\frac{a^2}{\beta_1^2\epsilon}\frac{\beta_1\beta_2\epsilon}{ab} + \frac{a^2}{\beta_1^2\epsilon}c\frac{\beta_1\beta_3\epsilon}{ac} \\
&= a + a\frac{\beta_2}{\beta_1} + a\frac{\beta_3}{\beta_1},
\end{aligned}
$$

and hence that

$$
\begin{aligned}
\frac{1}{K}\left(a + b\frac{K}{\sqrt{M}} + Kc\alpha^v\right)^2 &= \frac{a^2}{K}\left(1 + \frac{\beta_2}{\beta_1} + \frac{\beta_3}{\beta_1}\right)^2 \\
&= \frac{1}{K}\frac{a^2}{\beta_1^2}(\beta_1 + \beta_2 + \beta_3)^2 \\
&\leq \epsilon.
\end{aligned}
$$

Multiplying together th terms gives the second part of the result. $\square$

We can also show that this solution is not too sub-optimal.

**Theorem 27.** *Suppose that $a, b, c, \alpha > 0$. If $K, M, v > 0$ are set so that $\frac{1}{K}\left(a + b\frac{K}{\sqrt{M}} + Kc\alpha^v\right)^2 \leq \epsilon$, then*

$$
KMv \geq \frac{a^4b^2}{\epsilon^3}\frac{\log\frac{ac}{\epsilon}}{(-\log\alpha)}.
$$

*Proof.* The starting condition is equivalent to stating that

$$
\frac{a}{\sqrt{K}} + b\sqrt{\frac{K}{M}} + \sqrt{K}c\alpha^v \leq \sqrt{\epsilon}.
$$

Since all terms are positive, clearly each is less than $\sqrt{\epsilon}$. From this follows that

$$
\begin{aligned}
K &\geq \frac{a^2}{\epsilon} \\
M &\geq \frac{b^2 a^2}{\epsilon^2} \\
v &\geq \frac{\log \frac{ac}{\epsilon}}{(-\log \alpha)}.
\end{aligned}
$$

Multiplying these together gives the result. $\qquad\square$

**Theorem 28.** *If* $D \geq \max\left(\|\theta_0 - \theta^*\|_2, \frac{4R_2}{L}\log\frac{1}{\delta}\right)$, *then for all* $\epsilon$ *there is a setting of* $KMv$ *such that* $f\left(\frac{1}{K}\sum_{k=1}^K \theta_k\right) - f(\theta^*) \leq \epsilon_f$ *with probability* $1 - \delta$ *and*

$$
\begin{aligned}
KMv &\leq \frac{32LR_2^2 D^4}{\beta_1^4 \beta_2^2 \epsilon_f^3 (1-\alpha)} \log \frac{4DR_2 C}{\beta_1 \beta_3 \epsilon_f} \\
&= \mathcal{O}\left(\frac{LR_2^2 D^4}{\epsilon_f^3 (1-\alpha)} \log \frac{1}{\epsilon_f}\right) \\
&= \tilde{\mathcal{O}}\left(\frac{LR_2^2 D^4}{\epsilon_f^3 (1-\alpha)}\right).
\end{aligned}
$$

*Proof.* So, we apply this to the original theorem. Our settings are

$$
f\left(\frac{1}{K}\sum_{k=1}^K \theta_k\right) - f(\theta^*) \leq \frac{8R_2^2}{KL}\left(\frac{L\|\theta_0 - \theta^*\|_2}{4R_2} + \log\frac{1}{\delta} + \frac{K}{\sqrt{M}} + KC\alpha^v\right)^2.
$$

$$
\begin{aligned}
a &= \frac{L\|\theta_0 - \theta^*\|_2}{4R_2} + \log\frac{1}{\delta} \\
b &= 1 \\
c &= C \\
\epsilon &= \frac{\epsilon_f L}{8R_2^2}
\end{aligned}
$$

Note that, by the definition of $D$, $a \leq \frac{LD}{2R_2}$ and so $ac \leq \frac{LDC}{2R_2}$. Thus, the total amount of work is

$$
\begin{aligned}
KMv &= \frac{1}{\beta_1^4 \beta_2^2}\frac{a^4 b^2}{\epsilon^3}\frac{\log\frac{\beta_1\beta_3\epsilon}{ac}}{\log\alpha} \\
&= \frac{1}{\beta_1^4 \beta_2^2}\frac{a^4 b^2}{\epsilon^3}\frac{\log\frac{ac}{\beta_1\beta_3\epsilon}}{-\log\alpha} \\
&\leq \frac{1}{\beta_1^4 \beta_2^2}\frac{1}{\epsilon^3}\left(\frac{LD}{2R_2}\right)^4\frac{\log\frac{LDC}{\beta_1\beta_3 2R_2\epsilon}}{\log\alpha} \\
&= \frac{1}{\beta_1^4 \beta_2^2}\frac{8^3 R_2^6}{\epsilon_f^3 L^3}\left(\frac{LD}{2R_2}\right)^4\frac{\log\frac{LDC8R_2^2}{\beta_1\beta_3 2R_2\epsilon_f L}}{\log\alpha} \\
&= \frac{1}{\beta_1^4 \beta_2^2}\frac{32LD^4 R_2^2}{\epsilon_f^3}\frac{\log\frac{4DR_2 C}{\beta_1\beta_3\epsilon_f}}{\log\alpha} \\
&\leq \frac{32LD^4 R_2^2}{\beta_1^4 \beta_2^2 \epsilon^3 (1-\alpha)}\log\frac{4DR_2 C}{\beta_1\beta_3\epsilon}.
\end{aligned}
$$

$\qquad\square$

## 12 Strongly Convex Convergence

This section gives the main result for convergence this is true both only in the regularized case where $\lambda > 0$. Again, the main difficulty in this proof is showing that the sum of the norms of the errors of estimated gradients is small. This proof is relatively easier, as we simply bound all errors to be small with high probability, rather than jointly bounding the sum of errors.

**Lemma 29.** *With probability at least* $1 - \delta$,

$$\|e_{k+1}\|_2 \leq \frac{R_2}{\sqrt{M}}\left(1 + \sqrt{2\log\frac{1}{\delta}}\right) + 2R_2C\alpha^v$$

*Proof.* Once we have the difference of the distributions, we can go after the error in the gradient estimate. By definition,

$$\|e_{k+1}\|_2 = \|\frac{1}{M}\sum_{i=1}^{M}t(x_i^k) - \mathbb{E}_{p_{\theta_k}}[t(X)]\|_2$$

$$\leq \|\frac{1}{M}\sum_{i=1}^{M}t(x_i^k) - \mathbb{E}_{q_k}[t(X)]\|_2$$

$$+ \|\mathbb{E}_{q_k}[t(X)] - \mathbb{E}_{p_{\theta_k}}[t(X)]\|_2.$$

Consider the second term. We know by Lemma 17 and the assumption of mixing speed

$$\|\mathbb{E}_{q_k}[t(X)] - \mathbb{E}_{p_k}[t(X)]\|_2 \leq 2R_2\|q_k - p_k\|_{TV} \leq 2R_2C\alpha^v. \tag{6}$$

Now, consider the first term. We know that $\mathbb{E}_{q_k}[t(X)]$ is the expected value of $\frac{1}{M}\sum_{i=1}^{M}t(x_i^k)$. We also know that $\|t(x_i^k) - \mathbb{E}_{q_k}[t(X)]\| \leq 2R_2$. Thus, we can apply Theorem 15 to get that, with probability $1 - \delta$,

$$\left\|\frac{1}{M}\sum_{i=1}^{M}t\left(x_i^k\right) - \mathbb{E}_{q_k}[t(X)]\right\| \leq \frac{R_2}{\sqrt{M}}\left(1 + \sqrt{2\log\frac{1}{\delta}}\right). \tag{7}$$

Adding together Equations 6 and 7 gives the result. □

**Theorem 30.** *With probability at least* $1 - \delta$,

$$\|\theta_K - \theta^*\|_2 \leq (1 - \frac{\lambda}{L})^K\|\theta_0 - \theta^*\|_2 + \frac{L}{\lambda}\left(\sqrt{\frac{R_2}{2M}}\left(1 + \sqrt{2\log\frac{K}{\delta}}\right) + 2R_2C\alpha^v\right)$$

*Proof.* Apply the previous Lemma to bound bound on $\|e_{k+1}\|_2$ with probability at least $1 - \delta'$ where $\delta' = \delta/K$. Then, plug this into the main optimization result in Corollary 13. □

**Theorem 31.** *Suppose* $a, b, c > 0$. *Then for any* $K, M, v$ *such that* $\gamma^K a + \frac{b}{\sqrt{M}}\sqrt{\log\frac{K}{\delta}} + c\alpha^v \leq \epsilon$. *it must be the case that*

$$KMv \geq \frac{b^2}{\epsilon^2}\frac{\log\frac{a}{\epsilon}\log\frac{c}{\epsilon}}{(-\log\gamma)(-\log\alpha)}\log\left(\frac{\log\frac{a}{\epsilon}}{\delta(-\log\gamma)}\right)$$

*Proof.* Clearly, we must have that each term is at most $\epsilon$, yielding that

$$K \geq \frac{\log\frac{\epsilon}{a}}{\log\gamma}$$

$$M \geq \frac{b^2}{\epsilon^2}\log\frac{K}{\delta} \geq \frac{b^2}{\epsilon^2}\log\frac{\log\frac{\epsilon}{a}}{\delta\log\gamma}$$

$$v \geq \frac{\log(c/\epsilon)}{(-\log\alpha)}$$

From this we obtain that

$$KMv \geq \frac{b^2}{\epsilon^2} \frac{\log \frac{a}{\epsilon} \log(c/\epsilon)}{(-\log \gamma)(-\log \alpha)} \log\left(\frac{\log \frac{a}{\epsilon}}{\delta(-\log \gamma)}\right).$$

□

**Theorem 32.** *Suppose that $a, b, c, \alpha > 0$. If $\beta_1 + \beta_2 + \beta_3 = 1$, $\beta_i > 0$, then setting*

$$K = \log(\frac{a}{\beta_1 \epsilon})/(-\log \gamma)$$

$$M = \frac{b^2}{\epsilon^2 \beta_2^2}\left(1 + \sqrt{2\log \frac{K}{\delta}}\right)^2$$

$$v = \log\left(\frac{c}{\beta_3 \epsilon}\right)/(-\log \alpha)$$

*is sufficient to guarantee that $\gamma^K a + \frac{b}{\sqrt{M}}(1 + \sqrt{2\log\frac{K}{\delta}}) + c\alpha^v \leq \epsilon$ with a total work of at most*

$$KMV \leq \frac{b^2}{\epsilon^2 \beta_2^2} \frac{\log\left(\frac{a}{\beta_1 \epsilon}\right)\log\left(\frac{c}{\beta_3 \epsilon}\right)}{(-\log \gamma)(-\log \alpha)}\left(1 + \sqrt{2\log\frac{\log(\frac{a}{\beta_1 \epsilon})}{\delta(-\log \gamma)}}\right)^2$$

*Proof.* We define the errors so that

$$\gamma^K a = \epsilon\beta_1$$

$$\frac{b}{\sqrt{M}}(1 + \sqrt{2\log \frac{K}{\delta}}) = \epsilon\beta_2$$

$$c\alpha^v = \epsilon\beta_3.$$

Solving, we obtain that

$$K = \log(\frac{a}{\beta_1 \epsilon})/(-\log \gamma)$$

$$M = \frac{b^2}{\epsilon^2 \beta_2^2}\left(1 + \sqrt{2\log \frac{K}{\delta}}\right)^2$$

$$v = \log\left(\frac{c}{\beta_3 \epsilon}\right)/(-\log \alpha).$$

This yields that the final amount of work is

$$KMv \leq \frac{\log\left(\frac{a}{\beta_1 \epsilon}\right)\log\left(\frac{c}{\beta_3 \epsilon}\right)}{(-\log \gamma)(-\log \alpha)} \frac{b^2}{\epsilon^2 \beta_2^2}\left(1 + \sqrt{2\log\frac{\log(\frac{a}{\beta_1 \epsilon})}{\delta(-\log \gamma)}}\right)^2$$

□

*Remark* 33. For example, you might choose $\beta_2 = \frac{1}{2}, \beta_1 = \frac{1}{4}$ and $\beta_3 = \frac{1}{4}$, in which case the total amount of work is bounded by

$$KMv \leq \frac{4b^2}{\epsilon^2} \frac{\log\left(\frac{4a}{\epsilon}\right)\log\left(\frac{4c}{\epsilon}\right)}{(-\log \gamma)(-\log \alpha)}\left(1 + \sqrt{2\log\frac{\log(\frac{4a}{\epsilon})}{\delta(-\log \gamma)}}\right)^2$$

$$= \frac{4b^2}{\epsilon^2} \frac{\left(\log\left(\frac{a}{\epsilon}\right) + \log 4\right)\left(\log\left(\frac{4c}{\epsilon}\right) + \log 4\right)}{(-\log \gamma)(-\log \alpha)}\left(1 + \sqrt{2\log\frac{\log(\frac{a}{\epsilon}) + \log 4}{\delta(-\log \gamma)}}\right)^2$$

Or, if you choose $\beta_2 = 1/\sqrt{2}$ and $\beta_1 = \beta_3 = (1 - 1/\sqrt{2})/2 \approx 0.1464$, then you get the bound of

$$KMV \leq \frac{2b^2}{\epsilon^2} \frac{\left(\log\left(\frac{a}{\epsilon}\right) + 1.922\right)\left(\log\left(\frac{c}{\beta_3}\right) + 1.922\right)}{(-\log \gamma)(-\log \alpha)}\left(1 + \sqrt{2\log\frac{\log(\frac{a}{\epsilon}) + 1.922}{\delta(-\log \gamma)}}\right)^2$$

which is not too much worse than the lower-bound.

**Corollary 34.** *If we choose*

$$K \geq \frac{L}{\lambda} \log\left(\frac{\|\theta_0 - \theta\|_2}{\beta_1 \epsilon}\right)$$

$$M \geq \frac{L^2 R_2}{2\epsilon^2 \beta_2^2 \lambda^2} \left(1 + \sqrt{2\log\frac{K}{\delta}}\right)^2$$

$$v \geq \frac{1}{1-\alpha} \log\left(\frac{2LR_2 C}{\beta_3 \epsilon \lambda}\right)$$

*then $\|\theta_K - \theta^*\|_2 \leq \epsilon$ with probability at least $1 - \delta$, and the total amount of work is bounded by*

$$KMv \leq \frac{1}{\epsilon^2} \left(\frac{L}{\lambda}\right)^3 \frac{R_2}{2\beta_2^2(1-\alpha)} \log\left(\frac{\|\theta_0 - \theta\|_2}{\beta_1 \epsilon}\right) \left(1 + \sqrt{2\log\left(\frac{L}{\lambda\delta} \log\left(\frac{\|\theta_0 - \theta\|_2}{\beta_1 \epsilon}\right)\right)}\right)^2$$

*Proof.* Apply the previous convergence theory to our setting. We equate

$$(1-\frac{\lambda}{L})^K \|\theta_0 - \theta^*\|_2 + \frac{L}{\lambda}\left(\sqrt{\frac{R_2}{2M}}\left(1 + \sqrt{2\log\frac{K}{\delta}}\right) + 2R_2 C\alpha^v\right) = \gamma^K a + \frac{b}{\sqrt{M}}(1 + \sqrt{2\log\frac{K}{\delta}}) + c\alpha^v.$$

This requires the constants

$$\gamma = 1 - \frac{\lambda}{L}$$

$$a = \|\theta_0 - \theta\|_2$$

$$b = \frac{L}{\lambda}\sqrt{\frac{R_2}{2}}$$

$$c = 2LR_2 C/\lambda$$

Thus, we will make the choices

$$K = \log(\frac{a}{\beta_1 \epsilon})/(-\log\gamma)$$

$$\leq \log(\frac{\|\theta_0 - \theta\|_2}{\beta_1 \epsilon})/(1-\gamma)$$

$$= \frac{L}{\lambda} \log(\frac{\|\theta_0 - \theta\|_2}{\beta_1 \epsilon})$$

$$M = \frac{b^2}{\epsilon^2 \beta_2^2}\left(1 + \sqrt{2\log\frac{K}{\delta}}\right)^2$$

$$= \frac{L^2 R_2}{2\epsilon^2 \beta_2^2 \lambda^2}\left(1 + \sqrt{2\log\frac{K}{\delta}}\right)^2$$

$$v = \log\left(\frac{c}{\beta_3 \epsilon}\right)/(-\log\alpha)$$

$$= \log\left(\frac{2LR_2 C}{\beta_3 \epsilon \lambda}\right)/(-\log\alpha)$$

$$\leq \frac{1}{1-\alpha} \log\left(\frac{2LR_2 C}{\beta_3 \epsilon \lambda}\right)$$

This means a total amount of work of

$$KMv = = \frac{L}{\lambda}\log(\frac{\|\theta_0 - \theta\|_2}{\beta_1 \epsilon})\frac{L^2 R_2}{2\epsilon^2 \beta_2^2 \lambda^2(1-\alpha)}\left(1 + \sqrt{2\log\left(\frac{L}{\lambda\delta}\log\left(\frac{\|\theta_0 - \theta\|_2}{\beta_1 \epsilon}\right)\right)}\right)^2 \log\left(\frac{2LR_2 C}{\beta_3 \epsilon \lambda}\right)$$

$$= \frac{1}{\epsilon^2}\left(\frac{L}{\lambda}\right)^3 \frac{R_2}{2\beta_2^2(1-\alpha)} \log\left(\frac{\|\theta_0 - \theta\|_2}{\beta_1 \epsilon}\right)\left(1 + \sqrt{2\log\left(\frac{L}{\lambda\delta}\log\left(\frac{\|\theta_0 - \theta\|_2}{\beta_1 \epsilon}\right)\right)}\right)^2.$$

$\square$