[Reviews · NeurIPS 2015]

Submitted by Assigned_Reviewer_1

I think that the main issue of the manuscript is a lack of awareness of the literature on the topic. The authors' algorithm is strongly related to the Monte Carlo EM algorithm (and in fact the general class of stochastic approximation algorithms with markovian dynamic which have been proposed in various forms to optimize intractable likelihood functions). There is a well developed literature on the topic and the authors should compare what they have been doing to existing work. A theory which includes some of the authors' results already exists and covers much more general scenarios. For example the Markov chains are not required to be uniformly ergodic and there is no need to reinitialise the Markov chain at a fixed distribution r at each iteration (the authors acknowledge that they do this in order to simplify their analysis, but this is not sufficient--in addition reinitializing with the previous sample may be a better idea, as discussed at the end of the manuscript, and these algorithms have also been extensively analysed previously). The presentation of the results is far from optimal : for example for Theorem 1 and 2 it is not clear what the assumptions are? One can refer to the text before and after, but the reader should not have to do that. Note also that there is no need for the strong assumptions used in order to develop results such as those of Theorem 6 (see for example Benveniste, Metivier and Priouret for more general results).
Summary: I think that the authors lack awareness of the literature and the results are only marginally novel.

Submitted by Assigned_Reviewer_2

Summary: The paper proposes a novel approach to computationally efficient maximum likelihood learning in exponential families. In general, finding the maximum likelihood solution is intractable. From a convex optimization perspective, the sticking point is the need to calculate an integral wrt the currently proposed EF parameter. By assuming that MCMC is fast-mixing for all allowed parameters, the author(s) are able to show that the integrals needed for proximal gradient descent can be calculated with sufficient precision that, when combined with the results of Schmidt et al. (2011), a fully-polynomial randomized approximation scheme for calculating the MLE can be obtained. Both the convex and strongly convex cases are considered, which lead to different types of guarantees: the former on the likelihood error, the latter on the parameter error. A simple experiment demonstrating the theory is also included.

Overall, a high quality piece of work. The approach is well-motivated as an alternative to relying on small-treewidth assumptions. The theoretical analyses appear solid, though I haven't checked the proofs in detail. The proofs are essentially a matter of combining the MCMC mixing assumption with concentration inequalities and the results of Schmidt et al. The paper is thus a combination of existing techniques, though the combination is nontrivial.

I appreciated the honesty with which the limitations of the approach were discussed. In particular, a thoughtful conclusion analyzed weaknesses of the approach, ways in which the theoretical assumptions are overly pessimistic, and directions for future work. The paper also offers some insight into CD-type algorithms, even though the algorithm that is analyzed is not quite contrastive divergence.

The introduction was well-written with substantial references to previous work, but there were a few issues with the clarity of technical presentation. One small but important omission was simply the statement that the log-likelihood is in fact convex: the 2nd and 3rd terms in eq. 1 are clearly convex and the convexity of A(theta) follows from the fact that the hessian of A is the covariance of the sufficient statistics, and thus is PSD. While seemingly minor, the whole paper is premised on the the convexity of f(theta), and so discussing it will help guide the reader through the logic of the paper. A clear summary of the assumptions made in the paper. For example, one assumption never mentioned in the main paper is that the parameter set Theta must be convex.

Overall, I found the paper to be an original and substantial contribution. The results to seem to be of limited practical value due to the fact that proving fast mixing of MCMC is extremely challenging. So in a sense the paper has replaced one hard problem (finding the MLE) with another (finding fast-mixing Markov chain). That said, being able to convert one problem into another is often useful, and thus having the results of the paper available to the community is valuable.

Other comments: l185: ",w here" => ", where" l192: this paper that => this paper assumes that l207: simple example fast-mixing => simple example of a fast-mixing l281: of of => of l311-12: This sentence is nonsensical
Summary: An novel approach to developing conditions under which finding the (approximate) MLE is tractable. Well-written, though of seemingly limited practical value.

Submitted by Assigned_Reviewer_3

A key problem in Markov networks is weight learning, which is often defined as finding a setting of weights (attached to the features) such that the likelihood is maximized. Typically, a gradient descent algorithm is used to learn the weights and a key sub-task in this algorithm is computing the gradient. Since the latter is NP-hard in general, MCMC algorithms are often employed in lieu of exact algorithms to compute it.

The paper derives theoretical results showing that if the weights (parameters) are such that MCMC converges quickly to the stationary distribution on them, then the weight learning algorithm (that uses MCMC to compute the gradient) is a FPRAS. The paper considers two cases: unregularized case (convex case) and ridge-regularized case (non-convex case), and derives separate bounds for each.

The paper is generally well written with a few typos here and there. I was not able to verify all the proofs, however, they look plausible.

Although, the theoretical results look impressive, they are of little to no practical relevance (I like that the authors acknowledge this). I think it is virtually impossible to ensure that the parameters satisfy the constraints required by the theorems. Have the authors given this any thought? Specifically, whether it is possible to enforce this condition artificially (the condition can be seen as a form of regularization and actually using ridge regression in addition to it may yield over-regularized models). If this latter problem can be solved, together with these theoretical results, you have a very strong paper.

Overall, a reasonably well-written paper that formalizes the conditions under which using MCMC for weight learning (with and without regularization) yields a FPRAS. However, this condition is of little to no practical relevance. Non-trivial research problems need to be addressed before the results derived in the paper can yield a practical/accurate algorithm for weight learning.

Summary: A reasonably well-written paper that formalizes the conditions under which using MCMC for weight learning (with and without regularization) yields a FPRAS. However, this condition is of little to no practical relevance. Non-trivial research problems need to be addressed before the results derived in the paper can yield a practical/accurate algorithm for weight learning.

Submitted by Assigned_Reviewer_4

Comments on Quality:

The results give rates of convergence of learning algorithms in terms of mixing times. The bounds appear meaningful and useful.

The only significant complaint is that the lower bound (Theorem 7) is very difficult to interpret. An expanded discussion of the theorem and, more generally, the lower bounds applicable to the problem, would be desirable.

Comments on Clarity:

The presentation is clear.

Comments on Originality:

Both the framing of the problem (in terms of the "fast mixing" model class) and the theoretical results appear original.

Comments on Significance:

The significance is unclear to the reviewer. It does seem that the paper fills a gap in the literature on learning intractable models using MCMC, and in this sense it does seem significant.

The submission would be improved by further discussion of the work's potential impact (including future extensions and opportunities to build on it).
Summary: The submission introduces a novel class of tractable models based on good MCMC-based approximability of expectations. The theoretical results giving convergence rates for learning these models appear novel and interesting, and the submission should therefore be accepted.

Author Feedback
Author rebuttal: Thanks for all the detailed reviews, and the many helpful typos, clarity improvements.

On the impact of this work: The main impact is definitely theoretical. It is intuitive that max likelihood learning with MCMC should work given enough samples, a long enough and fast-mixing enough Markov chain, and enough gradient descent iterations. Nevertheless, this paper seems to be the first to give the type of high-probability convergence bound that it does, with specific settings of the parameters (step length, Markov chain length, number of gradient iterations, number of samples). Since learning using MCMC sampling is so common, it is important to verify that the intuition is indeed correct in this case, and what the parameter schedules achieve it.

While MCMC mixing times are difficult, reviewers take too pessimistic a view. Several cases where useful guarantees already exist: (1) MRFs with relatively weak interactions strengths as measured by a matrix norm (l 232), (2) Ising models with fixed degree and limited interaction strength depending on the degree (Mossel and Sly 2013 give a bound that is tight without extra assumptions) and (3) Ising models with arbitrary non-frustrated interactions and unidirectional field (l 224). Many other results exist in the mixing times literature. Due to different community priorities, these typically need some modification to be useful in machine learning, but this paper provides one motivation for this.

It is likely that many practitioners learning using MCMC will continue to set optimization parameters heuristically. However, even when doing so, insight might be gained from this analysis in terms of how the optimization parameters vary as a function of the acceptable error ep, and thus their relation to each other. (This analysis suggests K~1/ep, M~1/ep^2 and v~log(1/ep) in the convex case, and K~log(1/ep) when strongly convex.) Thus, this paper points an interesting direction to explore since these settings differ from common ones inspired by SGD. Again, however, this is speculative.

Finally, not every running algorithm has someone to carefully tune it. When, e.g., creating general-purpose software, it may be worth sacrificing speed for reliability. (Of course, this can only be done in a model class with a mixing-time guarantee.) The Chow-Liu algorithm remains useful, even though someone might often be able to engineer a system to learn a more complex model. We see this paper in this light.

R1

Please see the above on the difficulty of finding mixing time bounds.

R2

We will correct the extra page of citations by using a more compact style for them.

The proofs aren't based on any existing ones, other than the use of a large number of technical tools, mentioned in the sketches.

In Fig. 2, the left and center plots show 10 random runs of the algorithm, while the right plot shows one run, as compared to the ML parameters (far right). The hope here was to give a feeling for the variation in performance, since this is a randomized method/analysis.

Fig 1 appears formatted as intended, however any ideas to improve this would be welcome.

R3

Please see the above discussion trying to place this work in a larger context and discussing impact.

R4

We see Monte Carlo EM it as example of algorithms for learning using MCMC like those cited in Section 2. Absolutely, there is little *algorithmic* contribution in the paper, as similar algorithms (and elaborations thereof) have been used for a long time. The point that stochastic approximation can provide asymptotic results is well taken. Note that several papers in this line were cited [6,9,10] but more discussion is appropriate. However, we do believe this is the first non-asymptotic guarantee for an algorithm of this type, giving an exact running time to achieve a given error.

We will attempt to make the theorem statements more self-contained subject only to space constraints.

R5

The main practical weakness of this analysis is that it is conservative. (Like most theory for convex optimization.) So, for someone currently learning using MCMC with heuristic parameters, this paper offers only (1) the reassurance that the strategy isn't fundamentally unsound (2) possible intuition into how to set the different parameters to achieve a given error (3) a conservative way to set parameters to spend CPU to avoid parameter tuning. (Naturally, we feel the theory is important in its own right.)

We do disagree with the idea that mixing times are never known in practical cases. (All other parameters are easy to verify or bound.) See the above discussion on this. Also, note that a bound based on unknown parameters may still be useful- consider gradient descent bounds based on unknown Lipschitz conditions.

The "double-regularization" point is insightful. Sometimes mixing regularization much be such that the optimal bias/variance tradeoff might be made by lambda=0 (motivating the convex analysis).